# Exploring HIV risk perception mechanisms among youth in a test-and-treat trial in Kenya and Uganda

**Lawrence Owino**[1◉], **Jason Johnson-Peretz**[2◉]*, **Joi Lee**[2◉], **Monica Getahun**[2◉],
**Dana Coppock-Pector**[2], **Irene Maeri**[1], **Anjeline Onyango**[1], **Craig R. Cohen**[2], **Elizabeth A. Bukusi**[1], **Jane Kabami**[3], **James Ayieko**[1], **Maya Petersen**[4], **Moses R. Kamya**[3,5],
**Edwin Charlebois**[6], **Diane Havlir**[7], **Carol S. Camlin**[2,6]

**1** Kenya Medical Research Institute (KEMRI), Nairobi, Kenya, **2** Department of Obstetrics, Gynecology & Reproductive Sciences, University of California, San Francisco, San Francisco, California, United States of America, **3** Infectious Diseases Research Collaboration (IDRC), Kampala, Uganda, **4** Divisions of Biostatistics and Epidemiology, School of Public Health, University of California, Berkeley, Berkeley, California, United States of America, **5** Makerere University College of Health Sciences, Kampala, Uganda, **6** Center for AIDS Prevention Studies, University of California, San Francisco, San Francisco, California, United States of America, **7** HIV, Infectious Disease and Global Medicine, University of California, San Francisco, San Francisco, California, United States of America

◉ These authors contributed equally to this work.
* jason.johnson2@ucsf.edu

**Data Availability Statement:** Due to ethics and confidentiality restrictions around HIV and youth, we cannot make the data publicly available.

## Abstract

Understanding risk perception and risk-taking among youth can inform targeted prevention efforts. Using a health beliefs model-informed framework, we analysed 8 semi-structured, gender-specific focus group discussions with 93 youth 15–24 years old (48% male, 52% female), drawn from the SEARCH trial in rural Kenya and Uganda in 2017–2018, coinciding with the widespread introduction of PrEP. Highly connected social networks and widespread uptake of antiretrovirals shaped youth HIV risk perception. Amid conflicting information about HIV prevention methods, youth felt exposed to multiple HIV risk factors like the high prevalence of HIV, belief that people with HIV(PWH) purposefully infect others, dislike of condoms, and doubts about PrEP efficacy. Young women also reported minimal sexual autonomy in the context of economic disadvantages, the ubiquity of intergenerational and transactional sex, and peer pressure from other women to have many boyfriends. Young men likewise reported vulnerability to intergenerational sex, but also adopted a sexual conquest mentality. Comprehensive sexuality education and economic empowerment, through credible and trusted sources, may moderate risk-taking. Messaging should leverage youth's social networks to spread fact-based, gender- and age-appropriate information. PrEP should be offered alongside other reproductive health services to address both pregnancy concerns and reduce HIV risk.

## Introduction

HIV remains an ongoing global health concern. Among 38 million cases worldwide, sub-Saharan African (SSA) youth between the ages of 15 and 24 bear a disproportional disease burden

Requests for data can be made to the UCSF IRB via the IRB@ucsf.edu email address or: UCSF Human Research Protection Program, Box 1288 490 Illinois Street, Floor 6 San Francisco, CA 94143 (Use postal code 94158 for FedEx).

**Funding:** Research reported in this article was supported by the National Institutes of Health, NIAID under award number UM1AI068636, NCT #01864603 (AIDS Clinical Trial Group, Kuritzkes PI; SEARCH Supplement, Havlir PI) and in part by the Bill and Melinda Gates Foundation and Gilead Sciences. CAK was also supported by funding from the National Institute of Mental Health (K23 MH114760) and the Gilead Research Scholars Program in HIV.

**Competing interests:** The authors have declared that no competing interests exist.

[1]. In 2020, girls from that cohort made up six out of seven newly-diagnosed HIV infections worldwide [1]. Despite the high HIV incidence in this region and age group, some studies indicate youth have not perceived themselves to be at high risk of HIV acquisition, hindering HIV prevention efforts [2]. A contextual understanding of HIV risk perception, particularly among young sub-Saharan African women, is needed to address this aspect of the HIV preventive care continuum [3–5]. Such insight will provide a guide to health promotion and disease prevention programs globally.

In 2018, the Government of Kenya standardized and rolled out "test-and-treat" where interested individuals diagnosed with HIV immediately started on ART regardless of the CD4 count or viral load copies. At the same time the government began informational campaigns around Pre-Exposure Prophylaxis (PrEP) [6]. After the WHO recommended that HIV-negative individuals use ARVs for pre-exposure and post-exposure prophylaxis (i.e., PrEP and PEP) to prevent HIV infection in case of exposure in 2012, Uganda recommended the adoption of PrEP in its National HIV and AIDS Strategic Plan 2015–2019 [7, 8]. Kenya approved the use of PrEP beginning in 2015. Between May 2017 and August 2018, PrEP use increased twelve-fold in Kenya (from 1,425 people in 2017 to 17,466 in 2018), and by 2019 Kenya had the largest PrEP programme in Africa [9, 10].

Despite these positive developments, according to the UNAIDS data from 2018, the pace of HIV testing and linkage to care was too low (at 67%) to meet the 90-90-90 UNAIDS goals, especially among men. Recent work still indicates perceived low risk perceptions among youth in the contexts of prevention options uptake [11, 12]. While efforts to address some of the underlying factors behind this low uptake, such as increasing partner communication via HIV self-testing kits and the relationship between one's own sexual behaviour and partner trust are ongoing, they present an opportunity to sharpen the commonly used Health Beliefs Model through the lens of how social script models influence risk perceptions [13]. This is especially pertinent to disentangling the relationship between sexual behaviour, communication, and trust.

We sought to evaluate youth risk perceptions in the context of the Sustainable East Africa Research in Community Health (SEARCH) trial, a community-based intervention tailored to HIV prevention and elimination by offering PrEP to individuals at risk of HIV in rural Ugandan and Kenyan communities, engaged in universal treatment and streamlined care services [14, 15]. This trial saw a massive uptake of testing, moving from 57% of the residents having had an HIV test before to over 90% within the first year of the study [16]. Many people learned their HIV status for the first time, which led to increased disclosure and consequent fear of stigma from disclosure, but also magnified linkage to care. Further, PrEP was offered at the end of Phase 1, which aimed at using streamlined care delivery to measure the impact of early diagnosis and ART delivery. Any one of these rapid changes could affect youth HIV risk perception. Understanding HIV risk perceptions among youth in SSA requires consideration of this dynamic character. The research question this paper examines is, "How are youth risk perceptions shaped by (while existing within) the context of rapid changes in the HIV prevention landscape?"

Risk perceptions are dynamic and modifiable in nature. Policies which acknowledge this complexity are crucial to achieve the greatest possible reach. We focus specifically on the complexities of risk perception as they relate to youth in the context of the rollout of PrEP and Test and Treat, as unpacking the complexity of youth HIV risk perception in full is beyond the scope of this paper. Such perceptions easily vary by demographic and contextual factors like age, education, wealth, religion, generational change and crucially, the introduction of novel therapies and more effective medications [17].

The Health Beliefs Model (HBM) posits that specific modifying variables, including how peers create normative behavior around risk by following context-dependent 'social scripts', influence risk perceptions and consequent uptake of preventative methods [2]. Modifying variables include the availability and accessibility of evolving health technologies, such as HIV testing and antiretroviral therapy (ART), community perceptions linked to HIV care and prevention, and HIV awareness as defined by community prevalence and incidence. The presence or absence of these influences may thus also affect what factors youth associate with HIV risk. Among the changing health technologies, rapid changes in HIV prevention between 2015 and 2018 included the introduction of PrEP and official support for U = U messaging campaigns. (U = U messaging informs the public that people with undetectable viral loads cannot spread HIV, under the slogan, "undetectable equals untransmissible").

According to the HBM, the perceived threat of an illness, modifying variables, and cues to action all influence the likelihood of engaging in health-promoting behavior. For example, low perceived HIV risk decreases the likelihood of HIV testing and uptake of preventive measures [2]. The HBM has been applied to understanding risky behavior among youth in sub-Saharan African settings with varying results since the early 1990s [18]. However, the utility of HBM for HIV prevention behaviour could not be confirmed for condom use among students in late 1990s / early 2000s in South Africa [19]. Later, researchers there found that HBM predicts only about 25% of the acceptability of routine HIV testing among students, with *perceived benefits* having the greatest weight [20]. In contrast, Oyekale and Oyekale used the HBM to actively promote behaviour change among Nigerian youths in 2010. However, they did not find constructs in the HBM to predict behaviour. Instead, maturity, access to testing, family wealth, and educational background predicted a greater likelihood of changing behavior. Researchers applied the HBM to examine predictors of condom use among migrant construction workers in 2018 in Cameroon. They found *perceived barriers* overshadowed perceived benefits in predictive value, but HBM overall was *not* a significant predictor of consistent condom use [21]. Finally, a 2020 study in Malawi emphasized that more factors than risk perception are at play in PrEP interest among women [22].

Importantly, perceived benefits and barriers, in addition to perceived threat, are themselves subject to modifying variables like the influence of normative behaviour among peers, which changes based on age, gender, socio-economic status, and education. Multiple studies have borne this out. Gender norms have been found to influence risk perceptions for both women [22] and among men (in Ghana) [23]. Family connectedness competes with peer-exclusive networks to influence risk perception among youth in Ethiopia and KwaZulu-Natal [24, 25], even though peer networks among youth in Kenya and Malawi remain important for nearly all youth [26].

Assessing risk perceptions in the context of peer norms is important because in models of health behavior, especially those concerned with HIV [27], two factors are fundamental to the adoption of behavioral change: a) the perceived severity of the illness and b) the perceived susceptibility to that illness as mediated by peer norms. These two factors together form the 'perceived threat' in the four-part Health Belief Model (HBM) [27]. *Therefore, the most useful parts of the HBM model to focus on concern perceptions that are amenable to peer norms.* In both Nigeria and Cameroon, the interplay between perceived barriers and perceived access on the one hand, and socially mediated factors like age, wealth, and education on the other, highlight the importance of these peer-related variables.

Peer norms are often mediated by social scripts. Social scripts are models of social behaviour that allow people to perform certain activities near automatically, with a minimum of disruption because the models are both familiar and expected within a particular context [19, 20]. A subset of social scripts includes gender and sexual scripts [28, 29]. In these scripts, people

play out certain expected behaviours which conform to the expected norms for their gender. Sexual scripts often intersect with gendered scripts to prescribe different (though overlapping) sexual behaviours for men and women [28].

Social and sexual scripts can be inherited from parents and communities but they can also be mediated or transmitted by peers within one's age cohort. These scripts often underlie the dynamics which give rise to different risk scenarios for young men compared to young women, as noted in studies on risk behaviour [4, 30] and the societal expectations driving those differences. Crucially, social scripts are dynamic; when two or more actors are aware their scripts differ, as in intercultural exchanges, a new script can be negotiated or 'co-written' between two actors to smooth interactions and set common expectations [29]. While we wish to highlight the role that social scripts play in fostering perceptions of risk and informing the resultant behaviour among youth in our cohort, the purpose of this study is not to elucidate such scripts in full. Rather, this study seeks to build on the most replicable aspects of the HBM to examine youth perceptions of HIV risks and severity in the context of a rapid expansion of universal HIV testing and treatment in communities.

An earlier qualitative analysis by SEARCH investigators explored dimensions of PrEP uptake as influenced by mediating factors such as PrEP-related beliefs, HIV risk perceptions, and motivators for PrEP uptake and continuation [14]. This paper, informed by critiques of the HBM, differs from previous work by exploring how *modifying factors* influence HIV risk perceptions. Although the data is five years old, and despite having rolled out the HIV test-and-treat in Kenya in 2018, HIV incidence rate among youth remains significant. This study not only records a specific moment in time, but has the potential to continue to inform how researchers examine risk perceptions in rapidly changing health technology environments [1].

## Methods

We use data collected through a qualitative study that was nested in the larger study Sustainable East Africa Research in Community Health (SEARCH) trial, a population-based HIV test-and-treat trial implemented in Ugandan and Kenyan communities with adult HIV prevalence from 7.9% (SW Uganda) to between 14.3% and 26% (Migori, Kisumu, and Homa Bay counties, Kenya) [31–33]. These communities are situated in the rural areas populated by fishermen, farmers, traders, school-going youth, and transporters such as bodaboda (motorcycle) drivers. The communities are also characterised by social dynamics like early marriages and intimate partner violence.

The qualitative sub-study employed both in-depth interviews (IDIs) and focus group discussions (FGDs) to explore individual perceptions. Because this was a population-level rollout of PrEP, we focused on FGDs alone because they provided insight into communal dialogue and debate in real-time, in contrast to IDIs which recorded individual perceptions without this communal dynamic influencing the reports.

### Study design, sampling, and data collection

In 2016 and 2017, the SEARCH trial (NCT#01864603) conducted annual gender-specific focus group discussions (FGDs) in local languages to explore attitudes towards HIV, HIV knowledge, HIV risk perceptions, and interest in PrEP among youth ages 15–24 in the context of population-level PrEP implementation [34]. (See Table 1.) Community resident youth who participated in the FGD were purposively sampled for age, gender and having lived in the community for the past 6 months, through SEARCH-recruited Local Area Mobilizers (LAM) drawn from different social groups. The LAMs represented a range of youth peers drawn from

**Table 1. Focus group distribution by community and participant numbers.**

| Region | Community | FGD Type | Number of participants | Age Range |
|---|---|---|---|---|
| **Western Kenya** | Nyamrisra | Male youth | 8 | 15–24 |
| | | Female youth | 9 | 15–24 |
| | Sena | Male youth | 11 | 15–24 |
| | | Female youth | 10 | 15–24 |
| **S. Western Uganda** | Rugazi | Male youth | 14 | 15–24 |
| | | Female youth | 17 | 15–24 |
| **Eastern Uganda** | Bogono | Male youth | 12 | 15–24 |
| | Kameke | Female youth | 12 | 15–24 |

various settings such as fishermen and fish traders, youth in school and out of school, boda-boda drivers, and farmers; they were further balanced by religion, marital status, age, and gender. Recruitment to the FGDs began on 16 September 2016 and ended on 5 May 2017 (the day of the final FGD).

Each session comprised 8–17 participants. A total of 93 youth participated across all FGDs (48% male and 52% female). Each FGD lasted between 1.5 to 2.5 hours. FGDs were conducted with youth in five communities across the regions: two in Eastern Uganda, one larger FGD in Southwestern Uganda (where cultural norms dictate accompanying one another to social forums, leading to larger gatherings), and two in Kenya (see Table 1). The FGD guides assessed attitudes, beliefs, and social group norms related to PrEP, as well as target population insights into potential implementation strategies. We conducted the sessions in private settings within the community to ensure both confidentiality and access. Trained qualitative data collection researchers (two women and one man), who were experts in the local languages and who were familiar with these community settings [LO, IM, and AO] facilitated the sessions and participated in the analysis. Each FGD had one moderator and one field-note taker. FGDs were audio recorded, transcribed and translated into English for analysis.

## Data analysis

A team of six qualitative researchers who collected data in the field, a study manager, two research assistants, and the lead investigator analyzed data from the eight FGDs. We developed a coding framework after reviewing a subset of transcripts where all coders were assigned two similar transcripts for inductive coding [35]. Whilst coding, the team met regularly to review any questions, including data saturation. Group members were assigned both IDIs and FGDs to code using the same codebook. We coded using Dedoose software for beliefs, attitudes, HIV risk perceptions, experiences with PrEP, and community norms associated with HIV [36]. The analytic approach was informed by the Health Belief Model to explore core mechanisms behind 1) perceived risk of HIV infection; 2) perceived severity of HIV infection; 3) how perceptions of susceptibility and severity were linked to modifying factors; and 4) cues to action in relation to HIV. Gender stratification within the dynamics of these core mechanisms furthered deeper analysis, during which the salience of social scripts became apparent.

## Ethical approvals

The study was approved by the institutional review boards of Makerere University, the Uganda National Council of Science and Technology, the Kenya Medical Research Institute, and the University of California, San Francisco. Our study included participants below the age of 18 whose HIV status are unknown; these individuals did not need parental consent to participate

in these interviews as both the National AIDS and STI Control Programme (NASCOP) and Kenya Medical Research Institute (KEMRI) affirm that individuals aged $\geq$ 12 years can provide independent consent for research about STIs, including HIV, similar to the Uganda National Council for Science and Technology (UNCST). Youth in our study provided independent, written consent for their participation.

## Results

Tightly connected social networks and widespread uptake of antiretrovirals shaped youth HIV risk perception. We found complex and varied risk perceptions among participants showing that their health beliefs were influenced by their social ties in each aspect we studied. Below we decompose the category of *perceived threat* into perceptions of risk and perceptions of severity, before we describe the resulting interaction of perceived threat with youth-reported observations about *peer norms* as they influence the uptake of PrEP and other behaviours. Gendered differences in the dynamics of risk susceptibility were evident, while perceptions of the severity of HIV showed less distinct variations by gender, as noted below. We specify participants by their FGD participant number (e.g., "P5"), sex, community, and country.

### Perceived susceptibility to HIV

Youth reported four factors which they believe put them at higher risk of HIV: following peer-mediated sexual scripts, the use of alcohol, intergenerational sex (tied to economic insecurity), and the "malice" of PLWH who "knowingly spread" HIV.

**Performing gendered sexual scripts.** Both young men and women evinced an awareness that sexual scripts among their peers differ by gender. For men, these scripts entail displaying sexual conquest through engaging with multiple sex partners in a sort of competition with other males:

*"There is also competition among the male youths, like who has the highest number of female partners. And they also don't consider the use of the condoms. . ..."*

*(P4, Male, Nyamrisra, Kenya)*

*"Another factor is lust, in the event that you meet a lady you desired for a long time, you would not want to waste the chance–and this time you are at risk."*

*(P5, Male, Sena, Kenya)*

*"Some boys in our community can keep making advances on us girls, and sometimes apply such pressure that we end up giving in–that's how some of us get infected with HIV."*

*(P2, Female, Rugazi, Uganda)*

For these men, personal desire combines with inter-male social competition, leading them to take available sexual opportunities when presented to them, even if it means engaging in condomless sex. It can also mean pressuring young women into having sex that they may not be interested in. The young men thus recognize how following the expected norms for gaining status in their peer group puts them at risk for HIV, but they fail to recongise how those norms put women at risk as well.

In contrast, young women discussed parental involvement (or lack thereof) in sex education, and how they turn to their peers for additional scripts as the reason why young women have multiple sex partners.

*"It's true we girls do not listen to what our parents educate us about: they tell us never to engage ourselves in sexual activities, but still, we as young girls go ahead and involve ourselves in those bad acts and that's where we get infected with HIV."*

**(P9, Female, Rugazi, Uganda)**

*"We blame our friends. You find your friend has many boys she's sleeping with and she also wants you to have one, which can put you at risk of HIV."*

**(P16, Female, Rugazi, Uganda)**

*"We young girls have a lot of libido, whenever you see a boy you feel like having sex with him right away. . .(laughter)."*

**(P10, Female, Rugazi, Uganda)**

Thus, while young women expressed a susceptibility to male coercion, they also sought sexual encounters when encouraged by their peers. While coercion compromised their sense of self-efficacy, both male coercion and female peer pressure increased their risk of HIV infection. Young women recognise that a lack of education around sex and HIV protection from their parents leads them to listen more to their peers, who also may not have information on how to protect against HIV. The peer norm is to have multiple sex partners, not to have HIV-preventive sex.

**Alcohol use.** Both young men and women felt alcohol consumption potentially increased their risk of acquiring HIV, but their rationale differed by gender. Young men felt that they gain more courage to make sexual advances when using alcohol, which then compromises their ability of practicing protected sex.

*"Taking alcohol can increase one's risk of getting infected with HIV. When drunk, people tend to be courageous, thus I may get the courage to approach a girl and have unprotected sex–yet that girl might be positive."*

**(P6, Male, Sena, Kenya)**

*"Alcohol consumption among women?–When one takes alcohol, you become unconscious and you may be raped by an infected man, an act that you only realize very late when you become sober."*

**(P7, Female, Nyamrisra, Kenya)**

Drinking venues are places to socialise—not simply as venues to find sex partners, but places where youth associate with friends and tighten or maintain social networks. Young men and women agree to the fact that alcohol consumption pose a great risk of acquiring HIV as it makes men careless about using condoms while subjecting women to possibility of non-consensual sex. Women also face peer pressure from their girlfriends to have many lovers; but young women did not describe facing pressure from other young women to get drunk.

**Intergenerational and transactional sex.** Transactional sex is a predisposing risk factor for HIV. This is true for both men and women, as our informants pointed out.

*"Most girls including myself, we are always looking for working class men [i.e. who have jobs that earn them money] without wanting to first know his HIV status."*

**(P15, Female, Rugazi, Uganda)**

*"Most girls tend to be materialistic; they do not get satisfied with what the parents give them. Thus these girls end up running after old men looking for more stuff after sleeping with them, and it's a common practice that such men do not want to use condoms after giving you their money. Therefore, chances are high that those men will move around infecting young girls since they can't let go the money."*

**(P10, Female, Rugazi, Uganda)**

Transactional sex creates a power dynamic in which women's lower financial position and the struggle to attain economic satisfaction leads to the offer of unprotected sex. Poverty, in fact, is the driving motive for transactional relationships as both young men and women describe.

*"Poverty among families also contributes to the spread of the infection, as the young ladies may not get all the support from their parents and so they go 'round looking for money from every Tom, Dick and Harry in exchange for money without knowing the HIV status of these individuals."*

**(P7, Male, Nyamrisra, Kenya)**

*"Our parents do not have enough money for school fees, so men use that to their advantage by opting to help us and contribute part of the school fees, but on condition that you first sleep with them. Remember you would not know those men's status–thus infecting you with HIV"*

**(P17, Female, Rugazi, Uganda)**

Youth express an inability to meet their material needs because of their family's economic status. Young women place priority on completing school while downplaying the risk of getting infected by the people who are ready to finance their education but who demand sex in return. Young women thus strive after getting an education no matter what the cost, evincing a paradoxical pursuit of a long-term goal with a short-term plan, inadvertently placing them at more immediate risk of HIV.

Young men also feel at risk, acknowledging they aren't the only lovers a woman may have, but also because young men have less income than older men. This means that women may choose older men as sex partners for financial reasons but choose younger men who are their peers for other reasons.

*"For me, what I can say? As youths we sleep around with women and girls so much! We have girlfriends who are having sex with older men, and these are the same girls we are also dating– and we usually sleep with them without using condoms. As a result, we end up getting HIV."*

**(P6, Male, Bogono, Uganda)**

Multiple sex partners is common among both male and female youth. Participants already acknowledged that young women seek older partners to pay for school fees; here we see they also have sex partners of their own age.

*"What usually affects us is that we tend to admire what is in the females. We may know their current boyfriends but we don't shy away from it, and may want to snatch them from their boys so that we may possess them as ours. After we succeed, we also ignore the use of condoms, and we also don't know their HIV status–but still we engage in sexual intercourse with them."*

**(P6, Male, Nyamrisra, Kenya)**

For men, sexual conquest is a preeminent factor; the social push for sexual activity is so strong men ignore their own knowledge of HIV prevention measures simply to fulfil this goal.

These attitudes play into the theme of male rivalry among peers, as noted above, in addition to sexual desire. This rivalry, however, isn't between young men and other young men, but between young men who don't have money and older men who do. Because of the need for money to attract young women, young men also engage in intergenerational sex.

*"You find that around the lakeside, the aged tend to rejuvenate themselves to attract the youths as they believe that the youth still have strength. And . . . [the areas] around the lake are also occupied by people from various parts off the country and you also do not know the HIV status of these people. These older women also lure the young men with money and other lucre taking them for tours."*

*(P2, Male, Nyamrisra, Kenya)*

Financial empowerment looks key to fulfilling sexual needs. Young men, like young women, are also susceptible to financial insecurity, and therefore not only young women are at risk, but young men, too. Financial insecurity motivates both young women and young men to engage in intergenerational sex. For young women, the motive is to complete school a key indicator of moving through a particular life stage, while for young men, whose work opportunities may not require completing school, intergenerational sex provides them with money to get themselves established—and potentially attract a mate.

**Non-disclosure and purposeful transmission beliefs.**   Youth believe one reason why HIV infection rates remain high is because of purposeful transmission of HIV.

*"There are people who do work like the boda-boda cyclists; if he knows that he is HIV positive he will make sure he sleeps with the girls he rides with just to make sure he infects them with the virus. There are even teachers who are doing the same. That teacher who knows he is HIV positive will always want to sleep with young girls so he can infect them with the virus"*

*(P5, Male, Bogono, Uganda)*

*"Even if I am the one with HIV, I would try to go on giving it to others; I would rather put in all the money that I have to see that everyone else gets a copy, because I wouldn't want to die alone."*

*(P17, Female, Rugazi, Uganda)*

The settings for purposeful transmission which youth describe are those settings which youth experience: types of commonly available transportation, the schoolhouse (a central setting for this youth life-stage), and the social interactions which happen in those places (e.g., between bodaboda riders and their customers, or teachers and students). Youth believe that bodaboda drivers and teachers are in a position to exercise power over their younger (or less affluent) customers and students. It seems to be an open secret that teachers not only interact with their students academically but according to the young girls that interaction also goes beyond to sexual relationships. Youth, potentially from past experiences with people who hold power over them, also ascribe the motivation for purposeful transmission to the desire not to be alone, hinting that youth believe people with HIV feel lonely or are otherwise ostracized from others and might want to bring others into their situation.Thus, youth perceive that not only their own behaviour, but the settings they commonly encounter as part of their life stage and in which they are expected to mature, put them at risk.

In a context of high variability in HIV and HIV treatment literacy, some youth also believe that HIV treatment allows people both to conceal their status and infect others.

*"It's hard to identify those who are positive, some of them pick their drugs from distant clinics. Just like P6 said, some of them still want to infect others and they don't want to disclose their status."*

**(P5, Male, Sena, Kenya)**

*"It has greatly increased the spread of the virus as people say that even if they are taking the drugs, they are still healthy—unlike before when if one was infected, then they had to face a death penalty. It has really contributed to the secret of who is infected and who is not."*

**(P2, Male, Nyamrisra, Kenya)**

For these young men, ignorance about Undetectable = Untransmissible messaging is manifest; They express fear that these individuals can take advantage of their healthy-looking bodies to intentionally infect others. However, like the earlier excerpt about not dying alone, youth are aware of the stigma and potential loneliness that results from an HIV diagnosis. Youth seem to believe that once infected with HIV, the expected course of action is to conceal it. and they recognise this produces loneliness or a sense of separation from other people, which is counter to their own life stage script where lots of social interaction is important. Youth then relate these perceptions to the idea of purposeful transmission.

**Behaviours linked to perceived low susceptibility to HIV.**   Despite some ignorance around U = U, youth do know about PrEP, and that some people are using it effectively.

*"In my village, male youths use condoms and PrEP. Most of them are using PrEP because once you take it you are protected."*

**(P6, Male, Sena, Kenya)**

*"I am married and both of us use PrEP, and I think our life is okay. I don't have the fear of getting infected even though it is not only transmitted through sex—you may get HIV when you get involved in an accident. There is no way that I can get infected now—same to my wife."*

**(P5, Male, Sena, Kenya)**

Knowledge about HIV prevention methods seem to be common among the youth. They engage in open discussion with peers and partners about their sexual practices including the prevention methods used. It is in this context that condoms and PrEP are highly talked about and reported. Subsequently, some youth understand that HIV treatment (ARVs) does lower community susceptibility to HIV.

*"I think if one adheres well to ARVs and they are virally suppressed their chances of infecting others are reduced, unlike someone who is not on care."*

**(P6, Female, Sena, Kenya)**

The knowledge of U = U is well understood by some youth. The excerpt just quoted, in fact, followed the earlier quote about not wanting to die alone; this participant attempted to correct her peer's misunderstanding in the immediate context of the FGD itself, showing that even within one community a range of knowledge and beliefs exist around HIV treatment. For the more informed youth, they perceive that the availability of ARVs lowers their HIV

susceptibility because individuals on treatment have a minimal chance of transmitting the virus to others. Youth who understand U = U and who believe that people are usually on treatment feel at low risk of HIV.

Youth engage in open discussion with peers and partners about their sexual practices and the effects of both treatment and prevention methods on HIV susceptibility. It is in this context that condoms, PrEP, and ARVs are talked about and reported.

### Perceived severity of HIV

The perceived consequences of HIV influence perceptions of HIV severity among young people. The widespread availability of ARVs has influenced perceptions surrounding the severity of the diseases in both directions, from very severe to now quite manageable.

Even though most youth downplay the inevitable result of untreated HIV (i.e., death), the end result of unsuppressed HIV is well within their consciousness.

*"Most youths know that HIV is a deadly disease and it kills, and many of them fear it. Yes, like P8 said, there are youth who take [understand] HIV now to be like malaria. However, most youth know HIV to be a disease that kills."*

**(P5, Male, Bogono, Uganda)**

Some youth lived through an era where they saw people die from AIDS, and know that it is fatal if left untreated. Yet their common experience of people living fine with HIV today because of ARVs has led to a belief that HIV is not severe. Nevertheless, youth do perceive perception that HIV is severe on several fronts.

*"There are times when you look at people who are HIV positive and how they suffer, you feel so bad. Personally I take time and think about it; First, you can suffer so much when you are HIV positive, but also swallowing drugs every day is burdensome and worrisome. So, for me, there are times I talk to my fellow young people and some do buy in to my ideas, whereas others do ignore me,"*

**(P3, Male, Kameke, Uganda)**

In the communities, young people interact and meet those believed to be living with HIV. For these men, seeing the (social) suffering among infected individuals, further worsened by the burden of daily pills, elicits a sense of empathy or compassion. As a result, they create time to talk to their peers about the severity of living with HIV; expressing empathy through talking to peers is also a social script for men as they grow to adulthood and develop leadership skills.

Some youth, though, did not perceive HIV to be severe, as the availability of ARVs have made HIV comparable to other illnesses, lessening its impact to a manageable state.

*"Some people are scared of taking ARVs and they try to stay safe. Others take [or] treat it as a joke, they know that if they acquire HIV, the ARVs are available for them to take."*

**(P7, Male, Rugazi, Uganda)**

*"Some people compare it to malaria: you will just take the drugs and go on with your life. Talk such as 'so and so is positive' is no longer there because they look healthy. People also think that those who are positive look healthier than those who are still negative."*

**(P5, Male, Sena, Kenya)**

For these youth, HIV is comparable to malaria, which is commonly experienced; youth also believe that the availability of ARVs cushions HIV severity too, leading some youth treat it as a joke to deflect attention from their own behaviour. Downplaying the severity of HIV by joking that ARVs are available plays into the idea, mentioned next, that youth should enjoy this stage of their life (i.e., follow the script laid out by this life-stage) despite potential negative consequences of their behaviour.

## Constraints and factors motivating engagement in prevention

Youth reported several modifying factors which shape their perceptions of risk and severity, and influence what they consider to be viable prevention options. These factors included sexual norms, the social consequences of a chosen prevention method, the physical side effects of a prevention method, and a lack of parental engagement.

**Sexual norms.**   Earlier, youth described how sexual scripts place them at risk of HIV. These scripts are associated with risk perception, but apparently a perception of risk that is mediated by peer norms. These norms influence the uptake of prevention behaviour:

> *"At times one may be afraid of going to the hospital to pick condoms and I might not want to ask my friends because I don't want them to know am going to have sex, so I will just have to take the risk."*

> *(P1, Male, Sena, Kenya)*

> *"Some are not really concerned and are heard saying that 'you should enjoy your youth'."*

> *(P2, Female, Sena, Kenya)*

Sexual norms are not evenly spread among all youth. Some young men prioritise fear of being seen by friends in accessing prevention methods. For others, unprotected sex is excused as a norm that falls under the umbrella of enjoying their youth–*reinforcing a particular social script that some may doubt is wise to follow.*

Thus, behaving according to peer norms or the social expectations for young people can modify the perceived threat of illness. This is seen especially when peers encourage one another to fulfil the 'social requirements' of a life stage through various forms of sexual activity or abstinence to conform to those peer norms.

**Appraising social outcomes of different methods of risk reduction.**   Youth further perceive that risk reduction methods have social consequences.

> *"Some young ladies don't use condoms for fear of losing their partners, because if they suggest it, their boyfriend will claim that they don't trust them. . .though that might be true. . .laughter. So, they would rather not lose their boyfriend than force them to use condoms."*

> *(P6, Female, Sena, Kenya).*

The social consequences of risk reduction methods revolve around questions of trust and what is expected between partners in a sexual or romantic relationship. This may relate to the earlier-described financial situation of some women, who feel that keeping a boyfriend will allow them to achieve other goals like completing education, or they can also relate to a less often discussed desire on the part of young people to gain sexual experience.

*"Most girls always want to explore and know how different methods taste, for instance a girl can also say no to a condom wanting to taste the sweetness in live sex."*

**(P12, Female, Rugazi, Uganda)**

*"When a girl is to sleep with a man and she tries asking for condoms, the man will react in a threatening way. "How can I eat a sweet in its pack?!" So the girl will fear and end up allowing herself to sleep with him like that."*

**(P11, Female, Rugazi, Uganda)**

*"Interviewer: How does masturbation help around preventing HIV?*

*Participant 14: Whenever a girl gets feelings for sex but she's not sure of the man's HIV status she is intending to sleep with, then she can masturbate and feel fine."*

**(P14, Female, Rugazi, Uganda)**

These young women are aiming at becoming adults who are sexually experienced (or mature), and who know what they want. However, this social script falls into a context with gendered power dynamics, which girls must also negotiate as they mature. Sexual practices are dictated by certain social constructions and the outcome is heavily dependent on these constructions. Youth recognise that these socially-followed constraints affect their risk of acquiring HIV and they find alternative ways to address that.

In the same vein, peer-mediated rumours about side effects from PrEP and condoms influence youth appraisals of the benefits and drawbacks of these methods, especially among women. Rumors about PrEP side effects often stifled PrEP uptake. Unsurprisingly, when youth felt conflicted about their risk of HIV, the risk of using PrEP becomes more significant.

*"I have a feeling PrEP can even kill, because I have heard some of my colleagues talk about how it makes people weak. So I am imagining if a person is not that strong after taking the drugs, [they] might collapse and die."*

**(P14, Female, Rugazi, Uganda)**

Conversations with peers affects youth perceptions about the benefits and harms of prevention methods. Misinformation can easily spread among these highly connected networks.

Condom uptake was also marked by significant gender differences. Young women expressed perceived susceptibility to HIV due to their failure to use condoms, which itself was due to negative side effects from this method.

*"The condoms tend to have some effect after using them. They have some chemicals and scent that tend to upset the stomach around the abdomen area and you may feel like vomiting–but I don't know whether I'm the only one with the problem–and therefore most ladies do not like using it. It is the 'red sure' condom that is given cheaply at the hospital which has this effect."*

**(P4, Female, Nyamrisra, Kenya)**

Condom uptake is greatly dependent on user experience. Questions about condom quality at access sites played a role in uptake. Both issues translate into the possibility of not engaging in protected sex to avoid the unpleasant side effects. In this FGD context, the majority expressed concern about the methods and their side effects, and the informant illustrates how women do seek answers from other women about their experiences using methods like

condoms when the social context allows for these discussions, even through indirect discourse ("I don't know if I'm the only one. . ."). The resulting conversations then shape not simply health beliefs but health behaviours. However, despite these collectively constructed norms, personal experience–of unpleasant side effects, in this case–can over-rule them.

Finally, *a lack of parental engagement*, education, and adolescent rebellion were also modifying influences felt among the youth. Parents were blamed for inadequately exerting appropriate support to the youth.

*"It is still the same, our parents do not educate us on such things, they fear to tell us, and they think that since we are students then we should have learnt all that from school, which is not true. In our community you will not find a parent who can sit his or her daughter down and start teaching them how to protect themselves from getting HIV."*

**(P10, Female, Rugazi, Uganda)**

Young women recognise that a lack of education around sex and HIV protection from their parents, and its further absence at school leads them to listen more to their peers. These peers, as shown above, may or may not have information on how to protect oneself from HIV, but they nonetheless have a significant role in establishing the norms and expectations that youth negotiate in this life stage.

## Discussion

In this qualitative study conducted in the context of the SEARCH test and treat trial during the widespread offering of PrEP, we sought to explore the mechanisms of risk perception as framed in the construct of modifying factors from the Health Belief Model among young men and women ages 15 to 24 in rural Kenyan and Ugandan communities. Risk perception among youth at the time of PrEP introduction in East Africa was not well known, and this paper adds an understanding of a dynamically changing context through the lens of the HBM's identification of the influence of those modifying factors. Specifically, we chose to focus on a specific modifying factor, i.e. the dynamic character of social scripts, as youth took those up and navigated through them in the context of a high HIV prevalence setting.

Participants cited the influence of peers at every stage in the formation of their health beliefs, whether as creating susceptibility to HIV through the impact of their peers' behaviours or as exemplifying norms for our participants to follow. Our study and others [2, 37] suggest youth perceive high susceptibility *because of inconsistent condom use*, alcohol abuse, and intergenerational (transactional) sex *among their peers*. This means that our results demonstrate a *post-facto* judgement of risk, rather than an *a priori* assessment. This contrasts with Schaefer, who asserted that risk perception increases condom use [38]; we argue that lack of condom use either by oneself or among one's peers increases risk perception before it increases condom use (if, in fact, condom use does increase). A study conducted in Uganda similarly showed the major drivers impeding behaviour change include not simply libido, but importantly peer pressure [39], while another conducted in Ethiopia mentioned peer pressure to have sex as a mediating factor [24]. Intimacy in the sense of pleasure was prioritized (sweet in its pack), but in contrast to much earlier findings from other regions, we did not find evidence of foregoing the use of condoms in order to demonstrate either virility or intimacy in the sense of trust or commitment in this sample [40–43].

Perceived threat, however, varies based on the particular social dynamics of gender and poverty in rural communities and the barriers these features raise to behavioural change. The study findings highlight the perceived susceptibility to transactional sex for both men and

women. All youth voiced a desire for economic stability, which makes them vulnerable to older people who use wealth and the stability it promises in exchange for sexual favours. Young women, in particular, expressed little control over partner behaviors or felt pressured to engage in transactional sex. This perception has some epidemiological foundation [44, 45]. Similar experiences among young adults in Malawi [30], Tanzania [46], and South Africa [45] underscore the widespread lack of autonomy young women have when engaging in transactional sex even when they initiate such relationships [47]. In the context of widespread transactional sex, youth believe older men contribute to HIV transmission, creating the perceived dynamic of increased infection rates among both young women and young men due to such age-asymmetrical relationships [45]. Transactional sex, however, is nuanced and has its own peer norms and agentic behaviour, and both women and men do sometimes seek older partners who can pay for school fees or buy them nice things in order to be competitive with their peers; however, in the context of the interpersonal dynamics, women's sexual agency in particular can be limited inasmuch as their reception of these material benefits is dependent on particular forms of sexual compliance [45, 46]. Further, this phenomenon produces different perceptions among men and women about each gender's goals in pursuing sexual relationships, with men believing women are materialistic and women believing that men seek sex and will exchange material goods to obtain it. Together, the combination produces a feedback loop which raises the risk of HIV for everyone involved in this complex sexual economy. Ultimately, motivations to engage in transactional sex in our sample, as well as those from elsewhere on the continent, are driven by multiple, interpenetrating factors.

A persistent rumour youth report is belief that others engage in purposeful transmission of HIV, which youth tie to lack to disclosure. In contrast, participants in a study conducted in Botswana did not perceive high HIV severity because respondents believed people on ARVs do not necessarily increase risky sexual behaviors [48]. A study of HIV-positive MSM in the Southeastern and Midwestern US indicated that depressive symptoms and compulsive sexual behaviour were more highly associated with beliefs in intentional transmission of HIV [49]. Suggestively, sexual compulsivity was also expressed by multiple participants in our study. While both men and women sought multiple partners, the gender differences point to differing sexual scripts youth grapple with adopting or changing. Men reportedly engage in a supremacy battle of "who owns many sex partners' among themselves, in line with other studies in Ethiopia [50]. Young women, in contrast, encourage their peers to also take multiple lovers as a form of 'helping one another out,' and debate over whether to follow their libido or be more sexually selective, while at the same time desiring to gain more sexual experience or maturity. In our sample we did not find many young women who seemed to prioritise being in a steady relationship, so we cannot say that they prioritized (non-financially-motivated) relationship security over using condoms. This is perhaps an artefact of the age group of our sample, and that once older, more stable relationships better fulfill gendered expectations and scripts. Self-fulfilling sexual needs through masturbation was also mentioned, suggesting that partnered sexual practice is not the sole script in a youth's life. A high perceived impact of masturbation correlated with a high level of abstinence, and therefore a correspondingly lower risk of HIV in other contexts [51].

Similar to other studies on the continent, youth also accused parents of failing to offer appropriate guidance, not only abandoning informational support to other sources like school teachers, but also offering inadequate material support to the young woman, leading her to seek support elsewhere [52]. A Botswana study also found parents' misperceptions about their adolescent children's relationships to be associated with higher risk activities on the part of the adolescents [53]. Although nothing guarantees that youth will listen to their parents, the accusation that they do not provide any guidance at all highlights the possibility that parents don't

have a 'script' of their own to provide their children. They can provide neither behaviours to illustrate, nor health beliefs to pass on–perhaps trusting that teachers teach the necessary information. This is similar to the pre-intervention state in at least one intervention study conducted in South Africa, which demonstrated success in overcoming barriers on both the youth and parental sides of the equation [52].

Finally, some studies have raised the concern that normalizing HIV and ART can have the effect of lowering risk perceptions and consequent preventive behaviours–including testing behaviours [54]. In one study in South Africa, like in ours, the younger cohort feel HIV is a manageable illness today, even though they nonetheless fear getting a positive HIV test result. While that attitude can be interpreted as a contradiction, in the HBM it is explainable by the difference between risk perception and disease severity. While perception of HIV severity is low because it is treatable, risk perception can remain high, especially when the 'risk' extends beyond risk of the disease itself to involve the social consequences of living with the disease. Such consequences can include stigma and challenges to one's social relations (e.g. finding a marriage partner, disclosure, etc). While such social consequences should be considered within the category of 'severity', too often a clinical approach focuses only on how a disease affects the body, which is then taken up socially. Broadening a strict disease-focus to include the social scientific recognition that illness is a biopsychosocial phenomenon that extends beyond the mere biomedical diagnosis of a disease means that care must be taken to determine whether lay participants include the social consequences of a diagnosis within perceptions about either severity of a disease or the risks it presents to their (socially-embedded) lives. In our cohort, youth appear to associate social consequences with over-all risk perception (i.e. likelihood of acquiring the infection and what it means socially), rather than severity (i.e. degree to which it affects the body and care of the embodied self).

## Limitations

Our study has some limitations. Because the data was gathered from focus group discussions, participants may have been unwilling to 'own' their own behaviour, and instead spoke more generally about what they see their peers doing. However, this meant the participants were able to report on both their own and their peers' views in a local context. This provided a rich and in-depth understanding of their ways of life and the factors which inform their social context. Additionally, our focus in this paper has been on gendered social scripts, but gendered power norms, including within sexual relationships also undergo change. That topic has been explored among Ugandan adults in the early years of the HIV epidemic, but how youth receive these norms in the context of Test and Treat has been understudied [55].

## Recommendations

The diffusion of misinformation through social networks suggests that peer education strategies should leverage these channels to spread fact-based, targeted, and age-appropriate information and present alternative social scripts for youth sexuality; this extends a Nigerian study which found media was a major source of information in changing dysfunctional beliefs about HIV/AIDS [56, 57]. Comprehensive sexuality education through trusted and credible sources, either in school settings, clinical or community settings with or without parents present, or delivered by peers and which could moderate risk-taking among youth is critical. Such education should be clear that while HIV is treatable and that treatment can prevent its spread to others, nonetheless, having HIV doesn't make one's life any easier and its consequences are both physical and social. In this way, increasing stigma around people with HIV is avoided. PrEP should simultaneously be offered alongside health education to address prevailing

concerns about alarmingly high HIV risk among youth. Given successes elsewhere, separate interventions focusing on parental communication around sexual and reproductive health–including not only mothers and daughters, but also fathers and sons–would also be beneficial. While this targets parents, rather than adolescents, the effects are seen in both adults and adolescents.

Finally, to lower risk, employment and other wealth generating programmes must target youth specifically. We agree with other researchers that it is important that policy-makers recognize the transactional dynamics in sexual relationships among youth which limit their ability to negotiate sexual risk behaviours with their partners–not only in urban areas like Dar es-Salaam, but also in rural areas around Lake Victoria in Kenya and Uganda [46]. Thus economic empowerment programmes, especially for young women, but also for young men, could help attenuate the risk of intergenerational and financially-induced sex while promoting educational opportunities which further reduce objective HIV risk in this age demographic.

These study findings enrich the diverse perspectives of participants sharing similar geographical settings in rural East Africa during a time of rapid change in the HIV treatment and prevention landscape, and are therefore useful in understanding broader HIV risk perceptions among young people 15–24 years of age in the region.

## Supporting information

**S1 File. Qualitative code list.**
(DOCX)

**S2 File. Focus group discussion guides.**
(DOCX)

**S1 Checklist.**
(DOCX)

## Acknowledgments

We gratefully acknowledge Dr. Dickens Aduda of JOOUST for reviewing the primary concept, the Ministries of Health of Kenya and Uganda, the SEARCH research team, collaborators and advisory boards, and especially all communities and participants involved.

## Author Contributions

**Formal analysis:** Jason Johnson-Peretz, Joi Lee, Monica Getahun, Dana Coppock-Pector, Irene Maeri, Anjeline Onyango, Carol S. Camlin.

**Funding acquisition:** Diane Havlir.

**Investigation:** Irene Maeri, Craig R. Cohen, Elizabeth A. Bukusi, Jane Kabami, James Ayieko, Maya Petersen, Moses R. Kamya, Edwin Charlebois, Diane Havlir, Carol S. Camlin.

**Project administration:** Jason Johnson-Peretz, Maya Petersen.

**Supervision:** Jason Johnson-Peretz, Carol S. Camlin.

**Writing – original draft:** Joi Lee, Monica Getahun.

**Writing – review & editing:** Lawrence Owino, Jason Johnson-Peretz, Joi Lee, Carol S. Camlin.

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
