## [Decision Letter · Decision Letter 0]

27 Dec 2023

PGPH-D-23-02191

Exploring HIV risk perception mechanisms among youth in a test-and-treat trial in Kenya and Uganda

Dear Dr. Johnson Peretz,

Thank you for submitting your manuscript to PLOS Global Public Health. After careful consideration, we feel that it has merit but does not fully meet PLOS Global Public Health’s publication criteria as it currently stands. Therefore, we invite you to submit a revised version of the manuscript that addresses the points raised during the review process.

Editor comments:

The reviewers and I found the manuscript to be well-written and feel it provides valuable insights into how youth perceive risk in the context of test-and-treat in Kenya and Uganda.I agree with Reviewer 2 (see attached file with detailed comments) that some additional clarification in the methods would be helpful, and that aligning the write-up with COREQ or similar qualitative writing guidelines would improve the manuscript. I would also encourage the authors to include their final data collection and analysis tools (FGD guide, final codebook, etc.) in the supplemental materials.I would also encourage the authors to consider and respond to the questions raised by Reviewer 1, which may help clarify and position your findings in the context of the literature.

We look forward to receiving your revised manuscript.

Kind regards,

Marie A. Brault, PhD

Academic Editor

Journal Requirements:

2. Please include a complete copy of PLOS’ questionnaire on inclusivity in global research in your revised manuscript. Our policy for research in this area aims to improve transparency in the reporting of research performed outside of researchers’ own country or community. The policy applies to researchers who have travelled to a different country to conduct research, research with Indigenous populations or their lands, and research on cultural artefacts. The questionnaire can also be requested at the journal’s discretion for any other submissions, even if these conditions are not met.  Please find more information on the policy and a link to download a blank copy of the questionnaire here: https://journals.plos.org/globalpublichealth/s/best-practices-in-research-reporting. Please upload a completed version of your questionnaire as Supporting Information when you resubmit your manuscript.

4. In the online submission form, you indicated that "Data are available upon request. Due to ethics and confidentiality restrictions around HIV and youth, we cannot make the data publicly available". All PLOS journals now require all data underlying the findings described in their manuscript to be freely available to other researchers, either 1. In a public repository, 2. Within the manuscript itself, or 3. Uploaded as supplementary information.

Additional Editor Comments (if provided):

Reviewers' comments:

Reviewer's Responses to Questions

**Comments to the Author**

1. Does this manuscript meet PLOS Global Public Health’s publication criteria? Is the manuscript technically sound, and do the data support the conclusions? The manuscript must describe methodologically and ethically rigorous research with conclusions that are appropriately drawn based on the data presented.

Reviewer #1: Yes

Reviewer #2: Yes

2. Has the statistical analysis been performed appropriately and rigorously?

Reviewer #1: N/A

Reviewer #2: Yes

3. Have the authors made all data underlying the findings in their manuscript fully available (please refer to the Data Availability Statement at the start of the manuscript PDF file)?

Reviewer #1: Yes

Reviewer #2: No

4. Is the manuscript presented in an intelligible fashion and written in standard English?

Reviewer #1: Yes

Reviewer #2: Yes

5. Review Comments to the Author

Reviewer #1: Overall this paper is well written and addresses an important issue relating to young people’s HIV risk perception, a critical consideration. I have made a small number of specific comments, and then some more general comments for the authors’ consideration, as well as some recommendations and suggestions for further reading.

Specific comments

- Line 14: this the first mention of PrEP – so the acronym should be explained here at first mention

- Line 457: poses?

- Line 460: check punctuation

General comments

1) It would be interesting to unpack the complexity of risk perception around HIV. Prior research from sub Saharan Africa has shown that young people’s narratives of risk perception can be inconsistent and at times contradictory, suggesting that there are dualistic and contradictory notions of fear of HIV infection on the one hand – linked to the persistent and enduring association between HIV and death, and anticipated judgement, rejection and discrimination, while at the same time and a lack of concern regarding HIV infection linked to the destigmatisation and normalisation of HIV and ARVs. How should programmes and policies balance these 2 notions – while not increasing stigma around HIV, but at the same time ensuring that HIV infection is not something considered trivial.

2) The authors state that increased awareness of ARVs and reduces HIV stigma, young people also express less concern about getting infected with HIV. Prior research shows that perceptions of ARVs as efficacious and readily accessible can be linked to lower levels of concern of HIV infection. It is crucial to consider the extent to which ARVs have transformed perceptions of HIV from a life-threatening condition to a manageable, long term, chronic condition – and how this may have resulted in disinhibition or risk compensation behaviour.

3) I like the author’s use of scripting theories – a useful way to look at young men’s social competition, masculinities and gendered scripts. Other research has also shown that sexual prowess, proof of power over partners, and the prestige associated with condomless sex, and impregnating a female partner were are woven into gendered scripts for young men in sub-Saharan Africa. Young men prioritise prestige amongst peers, and demonstrations of their sexual maturity – at times proven by impregnating a girl. Condomless sex has also been associated with concepts of manhood and masculinity – regarded as more immediate and important than considerations around HIV infection.

4) Another critical theme that this paper addresses relates to SRH communication gaps between adolescents and parents. Evidence shows that where there are barriers to effective SRH communication between parents and adolescents, girls end up getting advice from peers and making poorly informed decisions. But it is not always the fault of parents – girls also may not want to listen to parents, regarding them as outdated and irrelevant – highlighting the generation gap.

5) It would be beneficial if the authors also consider the complexity around transactional sex and relationships. Is it as straightforward as poverty being the sole driving motive for transactional relationships? What about gender and peer norms – around fashion, clothing, material displays of wealth and ‘glamour’ in an effort to gain peer acceptance and social status? How about the agency of young women in choosing to use their sexuality as a currency? The authors suggest that there is a lack of autonomy when young women engage in transactional sex. However evidence suggests that young women may also be enacting some level of sexual agency and using their attractiveness and sexual ‘value’ to attain things they want – i.e. education, upward social mobility?

6) The authors state that for some young people, unprotected sex is excused as a norm that falls under the umbrella of enjoying their youth. Where does the prioritisation of pleasure play into this? Also, to what extent do young women prioritise relationship security over sexual health? Research amongst young people has shown that key factors determining condom use motivations for adolescent girls relate to relationship security and the desire to demonstrate love, trust, intimacy and commitment. In this manner, young women demonstrate a prioritization of romantic security and intimacy over their own sexual health, and seem to be willing to put themselves at risk for the sake of demonstrating trust, and prioritizing romantic connection. This can also be linked back to gendered scripting – and the framing of the importance of being in a relationship for young women = and the social prestige that brings. It has been shown that the self-esteem and social status of young women is often linked to their being in a romantic relationship, and therefore the security and maintenance of relationships is prioritized, which therefore can compromise risk avoidance behaviours.

7) Whilst gendered scripts are a critical consideration, it is also crucial to consider the way in which gendered power norms can also undergo shifts, and the ways in which young people can challenge them. Sexual scripts relating to gendered power are embedded in and informed by socio-cultural norms – which are themselves not static and unchanging – but are indeed slow to shift. Young people in sub-Saharan Africa navigate the duality between stubbornly persistent traditional values and expectations which co exist alongside more modern expectations.

Some suggested further reading relating toi the points above:

Legemate, E. M., Hontelez, J. A. C., Looman, C. W. N., & de Vlas, S. J. (2017). Behavioural disinhibition in the general population during the antiretroviral therapy roll-out in Sub-Saharan Africa: Systematic review and meta-analysis. Tropical Medicine & International Health, 22(7), 797–806. https://doi.org/10.1111/tmi.12885

Tariq, S., Hoffman, S., Ramjee, G., Mantell, J. E., Phillip, J. L., Blanchard, K., Lince-Deroche, N., & Exner, T. M. (2018). “I did not see a need to get tested before, everything was going well with my health”: A qualitative study of HIV-testing decision-making in KwaZulu-Natal, South Africa. AIDS Care, 30(1), 32–39. https://doi. org/10.1080/09540121.2017.1349277

Walker, L. (2020). Problematising the Discourse of “Post-AIDS.”. The Journal of Medical Humanities, 41, 95–105. https://doi.org/10.1007/ s10912-017-9433-9

Duby, Z., Jonas, K. McClinton Appollis, T., Maruping, K., Dietrich, J., Vanleeuw, L. & Mathews, C. (2020) “There is no fear in me … well, that little fear is there”: dualistic views towards HIV testing among South African adolescent girls and young women, African Journal of AIDS Research, 19(3). DOI: 10.2989/16085906.2020.1799232

Allen A. POWER TALK: YOUNG PEOPLE NEGOTIATING (HETERO)SEX. Women’s studies. Int Forum. 2003;26(3):235–44. doi https://doi.org/10.1016/S0277-5395(03)00053 – 0.

Duby, Z., Jonas, K. McClinton Appollis, T., Maruping, K., Dietrich, J. & Mathews, C. (2021) “Condoms Are Boring”: Navigating Relationship Dynamics, Gendered Power, and Motivations for Condomless Sex Amongst Adolescents and Young People in South Africa, International Journal of Sexual Health, 33(1). https://doi.org/10.1080/19317611.2020.1851334

Mantell JE, Needham SL, Smit JA, Hoffman S, Cebekhulu Q, Adams-Skinner J, Exner TM, Mabude Z, Beksinska M, Stein ZA, Milford C. Gender norms in South Africa: impli- cations for HIV and pregnancy prevention among afri- can and indian women students at a south african tertiary institution. Cult Health Sex. 20

---

## [Decision Letter · Decision Letter 1]

25 Mar 2024

Exploring HIV risk perception mechanisms among youth in a test-and-treat trial in Kenya and Uganda

PGPH-D-23-02191R1

Dear Mr Johnson Peretz,

We are pleased to inform you that your manuscript 'Exploring HIV risk perception mechanisms among youth in a test-and-treat trial in Kenya and Uganda' has been provisionally accepted for publication in PLOS Global Public Health.

Best regards,

Marie A. Brault, PhD

Academic Editor

Reviewer Comments (if any, and for reference):

Reviewer's Responses to Questions

**Comments to the Author**

1. If the authors have adequately addressed your comments raised in a previous round of review and you feel that this manuscript is now acceptable for publication, you may indicate that here to bypass the “Comments to the Author” section, enter your conflict of interest statement in the “Confidential to Editor” section, and submit your "Accept" recommendation.

Reviewer #1: All comments have been addressed

Reviewer #2: All comments have been addressed

2. Does this manuscript meet PLOS Global Public Health’s publication criteria? Is the manuscript technically sound, and do the data support the conclusions? The manuscript must describe methodologically and ethically rigorous research with conclusions that are appropriately drawn based on the data presented.

Reviewer #1: Yes

Reviewer #2: Yes

3. Has the statistical analysis been performed appropriately and rigorously?

Reviewer #1: N/A

Reviewer #2: Yes

4. Have the authors made all data underlying the findings in their manuscript fully available (please refer to the Data Availability Statement at the start of the manuscript PDF file)?

Reviewer #1: Yes

Reviewer #2: Yes

5. Is the manuscript presented in an intelligible fashion and written in standard English?

Reviewer #1: Yes

Reviewer #2: Yes

6. Review Comments to the Author

Reviewer #1: I am satisfied that the authors have addressed all my previous review comments, and believe the paper is now ready to be accepted. I have a few minor points for the authors' consideration, listed below.

Lines 177-178: data saturation is not a question – rephrase e.g. “review any questions and discuss data saturation”

Line 637: is “interpenetrating” the best term? Interlinked / intersecting?

Line 639: “rumour that youth report”

Reviewer #2: No additional comments to author.

7. PLOS authors have the option to publish the peer review history of their article (what does this mean?). If published, this will include your full peer review and any attached files.

**Do you want your identity to be public for this peer review?** For information about this choice, including consent withdrawal, please see our Privacy Policy.

Reviewer #1: No

Reviewer #2: No
